# Loc4Plan: Locating Before Planning
# for Outdoor Vision and Language Navigation

## ABSTRACT

Vision and Language Navigation (VLN) is a challenging task that requires agents to understand instructions and navigate to the destination in a visual environment. One of the key challenges in outdoor VLN is keeping track of which part of the instruction was completed. To alleviate this problem, previous works mainly focus on grounding the natural language to the visual input, but neglecting the crucial role of the agent's spatial position information in the grounding process. In this work, we first explore the substantial effect of spatial position locating on the grounding of outdoor VLN, drawing inspiration from human navigation. In real-world navigation scenarios, before planning a path to the destination, humans typically need to figure out their current location. This observation underscores the pivotal role of spatial localization in the navigation process. In this work, we introduce a novel framework, **Loc**ating be**for**e **Plan**ning (Loc4Plan), designed to incorporate spatial perception for action planning in outdoor VLN tasks. The main idea behind Loc4Plan is to perform the spatial localization before planning a decision action based on corresponding guidance, which comprises a block-aware spatial locating (BAL) module and a spatial-aware action planning (SAP) module. Specifically, to help the agent perceive its spatial location in the environment, we propose to learn a position predictor that measures how far the agent is from the next intersection for reflecting its position, which is achieved by the BAL module. After the locating process, we propose the SAP module to incorporate spatial information to ground the corresponding guidance and enhance the precision of action planning. Extensive experiments on the Touchdown and map2seq datasets show that the proposed Loc4Plan outperforms the SOTA methods.

## CCS CONCEPTS

• **Computing methodologies** → **Planning and scheduling**; • **Information systems** → *Multimedia information systems*.

## KEYWORDS

Vision and Language Navigation, Spatial Localization, Visual-textual Grounding, Cross-modal Matching

## 1 INTRODUCTION

Vision and Language Navigation (VLN) is a challenging task that requires agents to understand natural-language instructions and

Permission to make digital or hard copies of all or part of this work for personal or classroom use is granted without fee provided that copies are not made or distributed for profit or commercial advantage and that copies bear this notice and the full citation on the first page. Copyrights for components of this work owned by others than the author(s) must be honored. Abstracting with credit is permitted. To copy otherwise, or republish, to post on servers or to redistribute to lists, requires prior specific permission and/or a fee. Request permissions from permissions@acm.org.
*ACM MM, 2024, Melbourne, Australia*
© 2024 Copyright held by the owner/author(s). Publication rights licensed to ACM.
ACM ISBN 978-x-xxxx-xxxx-x/YY/MM
https://doi.org/10.1145/nnnnnnn.nnnnnnn

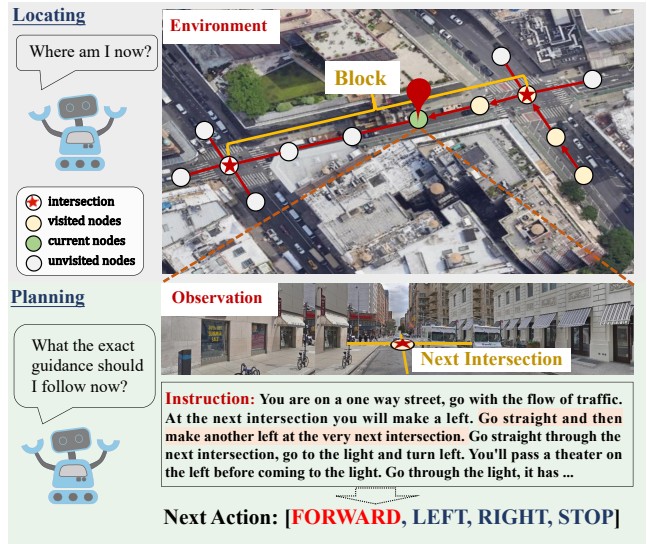

**Figure 1: The illustration of navigation process of our locating before planning approach. During the locating phase, the agent locates its relative spatial position in the current block. In the planning phrase, the agent associates the corresponding guidance to follow and makes an action decision to take (i.e., FORWARD).**

navigate to the destination in a visual environment. The agent is embodied in the environment and receives complete navigation instructions consisting of multiple sub-instructions to describe how to reach the destination step-by-step. Based on the instructions, the observed surroundings, and the current trajectory, the agent decides its next action. Executing this action changes the position and/or heading of the agent within the environment, and eventually, the agent follows the described route and stops at the desired goal location. Arguably, one of the key challenges in outdoor VLN is keeping track of which part of the instruction was completed. To alleviate this problem, various methods [25, 28, 37, 39] have been proposed. These methods mainly focus on grounding the natural language to the visual input, while neglecting the crucial role of the agent's spatial position information in the grounding process.

In this work, we argue that it is crucial for an agent to first determine its spatial position within the visual environment before grounding the appropriate guidance to follow in the outdoor VLN task. Consider a real-world navigation scenario for human, such as when a tourist seeks directions from a native in an unfamiliar area. The native typically begins by ascertaining the tourist's current spatial position before offering a route to the desired destination. This observation underscores the pivotal role of spatial localization in the navigation process. Unfortunately, previous studies have

overlooked the crucial importance of this spatial positioning stage, which significantly affects the agent's ability to interpret and execute navigation instructions accurately. Although the work [25] encodes some topological information, such as junction-type embedding and heading delta, to enhance agent generalization, it does not extensively explore the crucial role of spatial localization.

Drawing inspiration from human navigation, we first explore the effect of spatial position locating on the textual grounding of the outdoor VLN task. Generally, in human navigation, spatial positioning relies on prior knowledge of the navigated region's topology. However, outdoor VLN tasks usually require agents to navigate unseen environments, where the comprehensive environmental topology is unavailable during inference. Meanwhile, the visual observation perception of the agent is limited to a local region. To help the agent perceive its spatial location in the environment, we propose to learn a spatial predictor that measures how far the agent is from the next intersection for reflecting its position, which is achieved by a block-aware spatial locating (BAL) module. In our modeling, a "block" is defined as the area between adjacent intersections, as shown in Figure 1. In other words, each block represents a straight street segmented by two adjacent intersections. The BAL module enables the agent to determine its position at a finer granularity (block-level) rather than an intricate global-level, thereby facilitating the subsequent planning process.

The self-awareness of location ability developed in the BAL is beneficial for textual grounding, thereby facilitating further action planning. Therefore, we introduce the spatial-aware action planning (SAP) module, which incorporates the spatial locating information to associate the corresponding guidance and enhance the precision of action planning. In detail, we first identify the corresponding guidance that the agent needs to follow by associating spatial-aware state representation (obtained in BAL) with provided instructions in a hierarchical manner, ranging from sentence-level to token-level granularity. Specifically, the sentence-level association leverages the broader contextual understanding and richer semantics afforded by sentences. Subsequently, we devise a fine-grained mask derived from this sentence-level alignment to selectively filter out irrelevant information embedded in the token sequence. Compared to relying solely on word-level localization, our hierarchical semantic association provides a comprehensive understanding of the instructions, especially the extensive and intricate ones. Based on this identified corresponding guidance, the agent further incorporates spatial locating information into action decision planning.

Based on the above two modules, we construct a novel learning framework for addressing outdoor VLN tasks, named **Loc**ating be**for**e **Plan**ning (Loc4Plan), which enables agents to develop an ability of location-awareness like humans by first identifying the initial spatial localization before deciding where to go. Benefiting from the ahead localization to the agent's position and comprehensive understanding of the provided instructions, our Loc4Plan achieves the new state-of-the-art for Touchdown and map2seq dataset on seen and unseen scenarios, which outperforms ORAR framework[25] 3.3% and 4.8% of TC in *test unseen* scenario on the Touchdown [4] and map2seq [25] datasets, respectively.

In summary, the main contributions of this paper are as follows:

- We introduce a **Loc**ating be**for**e **Plan**ning (Loc4Plan) learning framework to address outdoor VLN tasks. Loc4Plan mimics the human navigation process by first determining the current location of the agent before making the next planning decision.
- To seek the location-awareness ability, we introduce the block concept in VLN tasks and propose a block-aware spatial locating (BAL) module to determine the agent's position within the given block, forming positional modeling.
- We introduce a spatial-aware action planning (SAP) module, which incorporates the spatial locating information to associate the corresponding guidance and enhance the precision of action planning.

## 2 RELATED WORKS

### 2.1 Vision-and-Language Navigation

The Vision-and-Language Navigation (VLN) task requires the agent to navigate in a 3D simulated environment toward the goal location based on instructions and egocentric observations. Various work has conducted in the indoor VLN task, including exploring feature representation [6, 11, 20, 22], modal interaction [1, 8, 30, 33], designing diverse mechanisms of reinforcement learning[1, 35, 36], scenic map building [5, 7, 18, 26], *etc.* Recently, research in the outdoor scenario has also begun to emerge and develop, which is based on real-world urban environments comprising actual street layouts and panoramic images [4]. Compared with indoor VLN, outdoor VLN contains a larger vocabulary and longer navigation instructions than indoor corpora [4], posing a greater challenge for agents to make cross-modal alignment between navigation state and long-span instruction. To solve this challenge, numerous methods have been proposed. Most of prior works [3, 4, 21, 25, 37] directly encode the observation, trajectory, and instructions in an LSTM-based model. The L2STOP [37] method differentiates STOP and other actions to boost the localization of stop action. GA [3] uses gated attention to compute a fused representation of instructions and images to predict actions. ORAR [25] adds junction-type embedding and a heading delta to improve the generalization of the agent in unseen scenarios. Additionally, many works[2, 15, 39] adopt the Transformer architecture for navigation, where VLN-Transformer [39] is the first transformer-based model of outdoor VLN. PM-VLN [2] method introduces the pre-training of priority map to achieve the temporal sequence alignment.

However, previous studies have overlooked the crucial importance of spatial positioning in navigation which significantly affects the agent's ability to interpret and execute navigation instructions accurately. In this study, we underscore the importance of initial spatial localization prior to planning decision actions. We introduce a novel framework called Locating before Planning (Loc4Plan) to tackle challenges in outdoor VLN tasks.

### 2.2 Textual Grounding in VLN

In vision-and-language navigation (VLN) tasks where the agent is provided with instructions detailing the entire navigation route, it is crucial to ground the linguistic guidance to the relevant segments pertaining to the next action. Consequently, developing effective techniques for textual grounding of step-wise navigation guidance

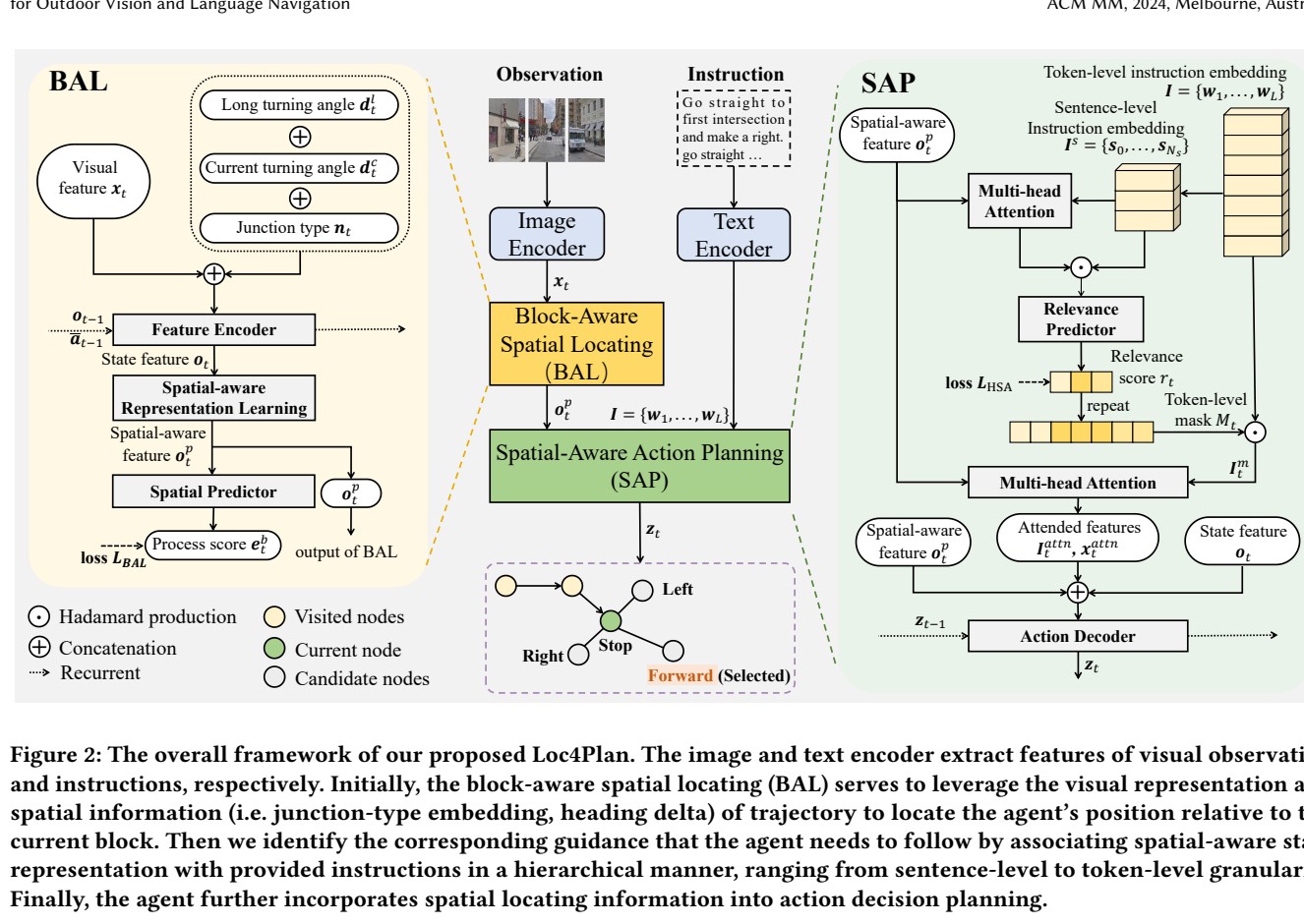

**Figure 2: The overall framework of our proposed Loc4Plan. The image and text encoder extract features of visual observation and instructions, respectively. Initially, the block-aware spatial locating (BAL) serves to leverage the visual representation and spatial information (i.e. junction-type embedding, heading delta) of trajectory to locate the agent's position relative to the current block. Then we identify the corresponding guidance that the agent needs to follow by associating spatial-aware state representation with provided instructions in a hierarchical manner, ranging from sentence-level to token-level granularity. Finally, the agent further incorporates spatial locating information into action decision planning.**

has emerged as a core challenge in VLN research. This challenge is not unique to VLN and represents an important issue that needs to be addressed for other multimodal tasks as well, as explored by various works[14, 16, 17, 27]. To achieve textual grounding, a main line of research [25, 28, 37, 39] employs cross-modal grounding over individual words between the natural language instruction and the environmental scene. And some of works[31, 39] implicitly make multi-modal alignment in an attention-based component of a transformer. Other works [2, 13, 15, 20, 33] focus on improving the representations of vision and language modalities, proposing auxiliary tasks to enhance grounding.

However, we find that relying solely on word-level localization for planning is insufficient, particularly when instructions are extensive. In this work, we align the observation and the instruction in a hierarchical manner, ranging from coarse to fine granularity.

## 3 LOCATING BEFORE PLANNING

### 3.1 Priliminary

**Problem Formulation.** Given a natural language instruction, a Vision-and-Language Navigation (VLN) agent is tasked with navigating from the starting position to the destination by following the guidance provided in the instruction. All navigation paths are instantiated on the directed graph environment $\mathcal{G} = (V, E)$ with

nodes $v \in V$ and labeled edges $(u, v) \in E$. Each node is associated with a $360°$ panorama image and each edge is labeled with an angle $\alpha_{u,v}$. At each timestep $t$, the agent's state $s_t \in \mathcal{S}$ is defined by $s_t = (v_t, \alpha_{(v_{t-1}, v_t)})$, where $v_t$ is the node at timestep $t$, and $\alpha_{(v_{t-1}, v_t)}$ is the heading angle from previous state's node $v_{t-1}$ to the current state's node $v_t$. Given the current navigation state $s_t$, the agent receives the corresponding visual observation from the environment. Based on the instruction information $\mathbf{I}$ and observed visual information $\mathbf{x}_t$, the agent infers the action $a_t$ from the candidate actions list of $[FORWARD, LEFT, RIGHT, STOP]$, and then executes the chosen action to update the next state $s_{t+1}$. The agent must produce a sequence of state-action pairs $[(s_1, a_1), (s_2, a_2), ..., (s_n, a_n)]$, where $a_n = STOP$, to reach the goal location.

**Block Definition.** We define a "block" as the region within the environment graph bounded by adjacent intersections, where traversal is constrained to a single path without the possibility of intersection crossings. An example of a block is shown in Figure.1.

### 3.2 Model Overview

Figure 2 provides an overview of our Locating before Planning (Loc4Plan) framework. Inspired by previous work [25], the model adopts a sequence-to-sequence architecture, which takes observational images and navigation instructions as input and outputs a sequence of agent actions. The Loc4Plan framework comprises a

block-aware spatial locating (BAL) module and a spatial-aware action planning (SAP) module. Specifically, the BAL module localizes the spatial position on the block level, which enables the agent to be aware of its relative location position within the current block. After this locating process, the SAP module leverages the spatial information (obtained in the BAL module) to identify the corresponding guidance that the agent needs to follow and make the action prediction. Further details of our Loc4Plan framework are presented below.

## 3.3 Block-Aware Spatial Locating

In human navigation, the initial stage entails identifying the user's current location before providing a routed path to the desired endpoint, which requires prior knowledge of the navigated region's topology. Drawing inspiration from human navigation, we integrate spatial localization into outdoor VLN tasks. However, navigating in outdoor VLN scenarios presents the challenge of maneuvering through unknown terrains where the complete environmental layout remains undisclosed. To address this, we introduce the block-aware spatial locating (BAL) module tailored for outdoor VLN, which establishes spatial positioning at a block-level granularity. We define blocks using intersections as demarcations, guaranteeing that the space between adjacent intersections pertains to the same block. Through the learning process facilitated by the BAL module, the agent gains an awareness of its current location, indicative of its relative position within the observation field at the block level.

Formally, consistent with the prior work [25], at timestep $t$ on node $v_t$, we incorporate the action embedding $\bar{\mathbf{a}}_{t-1}$ at timestep $t-1$, visual representation $\mathbf{x}_t$ of current observation, the turning angle of the current node $g_t^c$, and the junction type embedding $\mathbf{n}_t$ into a sequence modeling function to obtain the representation of the current state. Besides, we introduce the long-term turning angle $g_t^l$ as a novel input feature to model the long-range direction information. Then the state representation of the current node can be obtained by:

$$\mathbf{o}_t = \phi_1(\mathbf{o}_{t-1}, \bar{\mathbf{a}}_{t-1}, [\mathbf{x}_t \oplus g_t^c \oplus g_t^l \oplus \mathbf{n}_t]), \quad (1)$$

where $\phi_1$ is a feature encoder, $\oplus$ denotes the concatenation operation, $\mathbf{o}_{t-1}$ is the previous state of time step $t-1$, $\mathbf{n}_t$ is the embedding indicating the number of outgoing edges of the node at timestep $t$, and $g_t^c = Norm(\alpha_{(v_{t-1}, v_t)})$ is a value in $(-1, 1]$ that encodes the turning angle relative to the previous timestep. The long-term turning angle $g_t^l$ is computed by:

$$g_t^l = \sum_{k=0}^{K} g_{t-k}^c, \quad (2)$$

indicates the turning angle across consecutive steps, where $K$ denotes the number of steps involved in this calculation. When the special case that $t - k < 0$ occurs, we set $g_{t-k}^c = 0$. The concept of the long-term turning angle is introduced to aggregate turning angles across multiple consecutive steps, recognizing the process of turning left or right typically spans several individual actions.

The obtained node state presentation $\mathbf{o}_t$ has encoded information of previous states and observation visual and topological (e.g. junction-type embedding and heading delta) information of the current timestep. Further, we expect the agent can be aware of

the spatial position where they have achieved. To address this, we locate the relative position of the agent with the observation field under the block level. Specifically, we first obtain the spatial-aware state representation of the node $v_t$ by:

$$\mathbf{o}_t^p = ReLU(\mathbf{W}_b^\top \mathbf{o}_t), \quad (3)$$

where $\mathbf{W}_b \in \mathbb{R}^{d \times d}$ is a linear layer, and $\top$ indicates the transpose operation. We assume that the $\mathbf{o}_t^p$ accumulates relevant information beneficial for spatial localization.

**Optimization of BAL.** To ensure that spatial perception representation $\mathbf{o}_t^p$ can be aware of the information related to navigation progress within a block, we feed forward $\mathbf{o}_t^p$ to the spatial predictor to predict a navigation process score within current block by:

$$e_t^p = Sigmoid(\mathbf{W}_p^\top \mathbf{o}_t^p), \quad (4)$$

where $\mathbf{W}_p \in \mathbb{R}^{d \times 1}$ is a linear layer. $e_t^p \in [0, 1)$ indicates the agent's spatial position relative to the current block, where 0 indicates the start node of the block and 1 indicates the end. Block process score $e_t^p$ is supervised by the MSE loss:

$$L_{BAL} = \sum_{t}^{T} (e_t^p - \tilde{e}_t^p)^2, \quad (5)$$

where $T$ is the number of time steps to complete the entire navigation. And $\tilde{e}_t^p$ is the ground-truth block progress score at timestep $t$, which is calculated based on the number of nodes within the current block:

$$\tilde{e}_t^p = \frac{N_t^{step}}{N_t^{all}}. \quad (6)$$

where $N_t^{step}$ indicates the number of steps to "forward" to the next intersection node. And $N_t^{all}$ indicates the number of nodes in the current block (excluding the starting node). For example, when the agent is at the position shown in Figure 1, $N_t^{step} = 3$ and $N_t^{all} = 5$.

**Discussion.** With the learning of the BAL module, we assume the spatial-aware state representation has accumulated the information related to the spatial position. This suggests that the agent develops a self-awareness of location ability, just like humans.

## 3.4 Spatial-Aware Action Planning

The self-awareness of location ability developed in the BAL is beneficial for textual grounding, thereby facilitating further action plannin. Therefore, we introduce the spatial-aware action planning (SAP) module, which incorporates spatial locating information (i.e., the spatial-aware state representation obtained in the BAL module) to associate the corresponding guidance and enhance the precision of action planning. In detail, we first propose a hierarchical semantic association (HSA) submodule to identify the corresponding guidance that the agent needs to follow by associating spatial-aware state representation with provided instructions in a hierarchical manner, ranging from sentence-level to token-level granularity. We have found that relying solely on word-level localization is inadequate, especially when instructions are intricate and extensive. Consequently, we initially align spatial perception observations with instructions at the sentence level, leveraging the broader contextual understanding and richer semantics afforded by sentences. Subsequently, we devise a fine-grained mask derived from this

sentence-level alignment to selectively filter out irrelevant information within the instructions, thereby identifying the corresponding guidance for the current step. Based on this, the agent further incorporates spatial locating information into action decision planning.

**Hierarchical Semantic Association.** We first identify the corresponding guidance that the agent needs to follow by associating spatial-aware state representation with provided instructions in a hierarchical manner, ranging from coarse to fine granularity.

We start with associating the visual and textual in a sentence level. Specifically, given natural-language instructions $\mathbf{I} = \{\mathbf{w}_1, ..., \mathbf{w}_L\}$ with $L$ words, we use a period delimiter to split the instruction into multiple sentences and obtain the sentence-level embeddings $\mathbf{I}^s = \{\mathbf{s}_0, .., \mathbf{s}_{N_s}\} \in \mathbb{R}^{N_s \times d_t}$ by average pooling the token embedding of each sentence, where $N_s$ is the number of sentences. The sentence-level embedding $\mathbf{I}^s$ is then queried by the spatial-aware state representation $\mathbf{o}_t^p$ to model the sentence-level contextual feature $\hat{\mathbf{I}}^s \in \mathbb{R}^d$ by adopting multi-head cross-attention [34], which can be denoted as follows:

$$\hat{\mathbf{I}}^s = MultiHeadAttention(\mathbf{o}_t^p, \mathbf{I}^s), \quad (7)$$

where $\mathbf{o}_t^p$ serves as the *query* and $\mathbf{I}^s$ is utilized to calculate the *keys* and *values*. The sentence-level contextual feature contains coarsely-grained guidance relevant to the current state. Then, we utilize the sentence-level contextual feature $\hat{\mathbf{I}}^s$ as long as the sentence features $\mathbf{I}^s$ to calculate the relevance scores between the sentences and current state in the relevance predictor:

$$\mathbf{r}_t^{\hat{s}} = ReLU(\mathbf{W}_{\hat{s}}^\top \hat{\mathbf{I}}^s), \mathbf{r}_t^s = ReLU(\mathbf{W}_s^\top \mathbf{I}^s),$$
$$\mathbf{r}_t = Sigmoid(\mathbf{W}^\top (\mathbf{r}_t^{\hat{s}} \odot \mathbf{r}_t^s)), \quad (8)$$

where $\mathbf{W} \in \mathbb{R}^{d \times 1}$, $\mathbf{W}_{\hat{s}} \in \mathbb{R}^{d_t \times d}$, $\mathbf{W}_s \in \mathbb{R}^{d_t \times d}$, and $\odot$ denotes the Hadamard production. The relevance scores $\mathbf{r}_t = [r_{t,1}, ..., r_{t,N_s}] \in \mathbb{R}^{N_s}$ represents the possibility of each sentence being attended at timestep $t$.

Subsequently, we devise a fine-grained mask derived from this coarsely-grained relevance scores, which is utilized to filter out irrelevant information in the token embedding for fine-grained instruction attention. Specifically, for the $i$-th sentence, we obtain its corresponding token level mask $\mathbf{m}_{t,i} = [r_{t,i}, ..., r_{t,i}]$ by repeating the sentence-level relevance score $r_{t,i}$ with $L_i^s$ times, where $L_i^s$ denotes the length of the $i$-th sentence. Then the complete mask of the instruction can be denoted as $\mathbf{M}_t = [\mathbf{m}_{t,i}, ..., \mathbf{m}_{t,N_s}]$. We utilize the token mask $\mathbf{M}_t$ to filter out the irrelevant information in the instruction by:

$$\mathbf{I}_t^m = \mathbf{I} \odot \mathbf{M}_t, \quad (9)$$

Then the masked instruction $\mathbf{I}_t^m$ is used to obtain the attended instruction feature with the spatial-aware state representation $\mathbf{o}_t^p$ as query:

$$\mathbf{I}_t^{attn} = MultiHeadAttention(\mathbf{o}_t^p, \mathbf{I}_t^m). \quad (10)$$

Then we utilize the $\mathbf{I}_t^{attn}$ as query to obtain the attended visual feature:

$$\mathbf{x}_t^{attn} = MultiHeadAttention(\mathbf{I}_t^{attn}, \mathbf{x}_t), \quad (11)$$

**Action Planning.** With the attended visual representation $\mathbf{x}_t^{attn}$, attended textual representation $\mathbf{I}_t^{attn}$, state representation $\mathbf{o}_t$, as long as spatial-aware state representation $\mathbf{o}_t^p$, we feed-forward the

concatenation of these representations to the action decoder $\phi_2(.)$ to obtain the final representation for the action decision:

$$\mathbf{z}_t = \phi_2(\mathbf{z}_{t-1}, [\mathbf{x}_t^{attn} \oplus \mathbf{I}_t^{attn} \oplus \mathbf{o}_t \oplus \mathbf{o}_t^p \oplus \bar{\mathbf{t}}]), \quad (12)$$

where $\bar{\mathbf{t}}$ is the embedded timestep $t$. $\mathbf{z}_t$ contains cross-modal information for action planning, which is then integrated with turning angles $g_t^a$ of each action to predict action scores for the next step:

$$c_t^a = \mathbf{W}_a^\top [\mathbf{z}_t \oplus g_t^a], \quad (13)$$

where $\mathbf{W}^a \in \mathbb{R}^{d \times 1}$ is a linear layer. We define $g_t^a$ to represent the turning angles obtained if the agent executes action $action \in [FORWARD, LEFT, RIGHT, STOP]$ at timestep $t$. After the calculation of action scores for all candidate actions, the agent will execute the corresponding action of the maximum score and update the navigation state.

**Optimization of SAP.** The optimization of SAP comprises the learning of the hierarchical semantic association and the action planning loss. For the hierarchical semantic association, the sentence level relevance score $\mathbf{r}_t$ (Eq. (8)) is learned under the supervision of the sentence-level corresponding labels, which indicates the alignment of the sentences and nodes along the trajectory. Specifically, by referring to the multi-label classification method [32], a binary cross-entropy loss is used to supervise each sentence:

$$L_{HSA} = -\gamma_b \sum_t^T \sum_i^{N_s} (\tilde{r}_{t,i} log(r_{t,i}) + (1 - \tilde{r}_{t,i}) log(1 - r_{t,i})), \quad (14)$$

where $\tilde{r}_{t,i}$ is a binary label indicating whether the agent needs to attend to the $i$-th sentence when navigating to the node $v_t$. This label is generated from an economy-efficient method of template matching[12, 38]. $\gamma_b$ is the weight related to the confidence of the corresponding pseudo-label, which is used to control the influence of $L_{HSA}$ on training. More details can be found in Supplementary.

The action planning loss $L_{AP}$ is a regular cross-entropy loss:

$$L_{AP} = -\sum_t^T \sum_i^4 \tilde{c}_t^{a_i} log(c_t^{a_i}), \quad (15)$$

where $\tilde{c}_t^{a_i}$ denotes the binary ground-truth label indicating whether execute action $a_i$ at time step $t$, and $i$ indicates the action index of $[FORWARD, LEFT, RIGHT, STOP]$.

**Discussion.** By involving the spatial location information into the learning of the SAP module, we can ground the instruction guidance that the agent should adhere to in the current stage and facilitate action prediction.

## 3.5 The Overall Training

As mentioned above, our proposed Loc4Plan is optimized with three losses, i.e., $L_{AP}$, $L_{BAL}$, and $L_{HSA}$. The overall loss function can be written as:

$$L = L_{AP} + L_{BAL} + L_{HSA}. \quad (16)$$

During training, we use Teacher-Forcing [19] strategy to achieve stable optimization.

**Table 1: Comparisons with state-of-the-arts on the Touchdown and map2seq datasets for the seen and unseen scenario. The * indicates that the agent requires extra ground truth path traves during inference.**

| | Seen | | | | | | | | Unseen | | | | | | | |
| | Touchdown | | | | map2seq | | | | Touchdown | | | | map2seq | | | |
| | dev | | test | | dev | | test | | dev | | test | | dev | | test | |
| Model | TC↑ | SPD↓ | TC↑ | SPD↓ | TC↑ | SPD↓ | TC↑ | SPD↓ | TC↑ | SPD↓ | TC↑ | SPD↓ | TC↑ | SPD↓ | TC↑ | SPD↓ |
|---|---|---|---|---|---|---|---|---|---|---|---|---|---|---|---|---|
| GA[3, 4] | 9.9 | 21.4 | 9.7 | 21.5 | 8.1 | 39.1 | 7.3 | 40.7 | 2.5 | 30.1 | 1.8 | 30.2 | 1.1 | 45.4 | 0.5 | 45.6 |
| RCONCAT[4, 21] | 11.1 | 19.9 | 9.7 | 21.7 | 11.0 | 38.2 | 7.3 | 39.5 | 3.6 | 29.1 | 2.4 | 29.1 | 1.5 | 44.7 | 0.6 | 45.6 |
| ARC+L2STOP[37] | 19.5 | 17.1 | 16.7 | 18.8 | - | - | - | - | - | - | - | - | - | - | - | - |
| VLN Transformer[39] | 14.0 | 21.5 | 14.9 | 21.2 | 18.6 | 18.6 | 17.0 | 19.0 | 2.3 | 29.5 | 3.1 | 29.6 | 3.6 | - | 3.5 | - |
| ORAR[25] | 29.9 | 11.1 | 29.1 | 11.7 | 43.4 | 7.2 | 41.7 | 7.6 | 15.4 | 20.0 | 14.9 | 20.7 | 27.6 | 11.9 | 30.3 | 12.7 |
| PM-VLN*[2] | 33.0 | 23.6 | 33.4 | 23.8 | - | - | - | - | - | - | - | - | - | - | - | - |
| Ours | 34.5 | 10.5 | 32.9 | 11.5 | 48.0 | 7.0 | 45.3 | 7.2 | 20.6 | 20.0 | 18.2 | 19.9 | 32.8 | 10.0 | 35.1 | 10.5 |

## 4 EXPERIMENTS

### 4.1 Experimental Setup

**Datasets**. We conduct experiments on two popular outdoor VLN datasets, i.e, the Touchdown [4] and the map2seq [24] datasets. The Touchdown dataset [4] is built based on Google Street View in real urban environments of New York City, whose environment graph includes 29,641 panoramas and 61,319 edges. It contains 9,326 paired samples of English instructions and trajectory descriptions, where the samples in train and test sets are mixed in terms of their geographic locations (i.e. seen scenarios). The unseen scenarios are then conducted by [25] that makes geographic separation of the training and testing area. The map2seq dataset [24] was created for the task of navigation instructions generation and introduced in outdoor VLN [25]. It is also constructed in New York environment, containing 7,672 samples with both seen and unseen scenarios.

**Evaluation Metrics**. Following [2, 4, 15], we adopt Task Completion (TC), Shortest-path Distance (SPD), and Success weighted by Edit Distance (SED) to evaluate the models. Specifically, TC represents the percentage of successful agent navigation. SPD measures the shortest path distance from the node the agent stopped to the goal node within the environment graph. SED represents the task completion weighted by the Levenshtein edit distance between prediction and ground-truth trajectories.

**Implementation Details**. We use a bidirectional LSTM [9] as the text encoder to extract token-level embedding. A ResNet pretrained on ImageNet [23] is utilized as image encoder to extract visual representation $x_t$. Following [25, 29, 37], we implement $\phi_1(.)$ and $\phi_2(.)$ as LSTMs [10]. We set the dimension of token embedding $d_t = 512$ and the hidden dimension of model $d = 256$. The dimension of timestep embedding is 32, and the size of action embedding and junction type embedding is 16. For the specific process of generating sentence-level label $\tilde{r}_{t,i}$ (Eq. (14)) and other training details, please refer to the Supplementary.

### 4.2 Comparisons with SOTA VLN Methods

In this section, we compare our proposed locating before planning (Loc4Plan) framework with previous state-of-the-art (SOTA) approaches [2–4, 21, 25, 39]. Except for ORAR [25], these methods only report results for the seen scenario on Touchdown. To enable a comprehensive comparison, we evaluate methods with published code, supplementing results on the map2seq dataset and

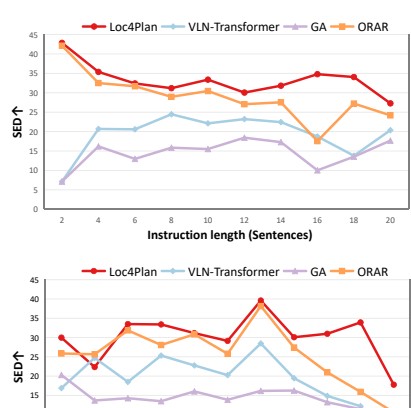

**Figure 3: Performance comparison with different instruction lengths and trajectory complexity.**

the unseen scenario of Touchdown. The results, as illustrated in Table 1, indicate that our method exhibits superior performance on both Touchdown and map2seq datasets. Remarkably, on the challenging *test unseen* split, our model showcases significant improvements over ORAR [25] by 3.3% on the Touchdown dataset and 4.8% on the map2seq dataset. This demonstrates the effectiveness of our Loc4Plan framework. In the *test seen* split of Touchdown, our Loc4Plan achieves performance comparable to that of PM-VLN, despite PM-VLN relying on additional ground-truth trajectory information for trajectory planning, which is not necessary for our Loc4Plan. Furthermore, our agent significantly enhances the SPD score by 12.3% compared to PM-VLN, underscoring its superior ability to adhere to instructions while navigating accurately.

Furthermore, we investigate the performance of different agents under varying trajectory complexities and instruction lengths, as illustrated in Figure 3. Specifically, we utilize the SED score to evaluate the navigation capabilities of these agents on the test set of the Touchdown dataset. Trajectory complexity is quantified by the number of intersections, while instruction length serves as a proxy for the level of difficulty in comprehension. As depicted, the Loc4Plan agent exhibits significant improvements, particularly

**Table 2: Ablation study of overall design on the test set in seen scenario.**

| ID | Models | Touchdown | | | map2seq | | |
|----|--------|-----------|---|---|---------|---|---|
| | | TC↑ | SPD↓ | SED↑ | TC↑ | SPD↓ | SED↑ |
| 1 | Baseline | 22.6 | 12.3 | 22.1 | 34.6 | 10.0 | 33.8 |
| 2 | With BAL | 30.3 | **11.0** | 29.6 | 39.0 | 8.7 | 38.4 |
| 3 | With SAP | 29.7 | 11.5 | 29.0 | 39.0 | 7.5 | 38.2 |
| 4 | Full model | **32.9** | 11.5 | **32.1** | **45.3** | 7.2 | **44.3** |

**Table 3: Ablation studies of detailed design of the block-aware spatial locating on test set in seen scenario..**

| ID | $g_t^l$ | $g_t^c$ | $L_{BAL}$ | $L_{GAL}$ | Touchdown | | | map2seq | | |
|----|---------|---------|-----------|-----------|-----------|---|---|---------|---|---|
| | | | | | TC↑ | SPD↓ | SED↑ | TC↑ | SPD↓ | SED↑ |
| 1 | | √ | √ | | 30.0 | 11.7 | 29.3 | 41.8 | **7.0** | 41.1 |
| 2 | √ | | √ | | 30.3 | 11.6 | 29.6 | 42.0 | 7.3 | 41.1 |
| 3 | √ | √ | | | 30.3 | 11.6 | 29.7 | 41.0 | 7.8 | 40.2 |
| 4 | √ | √ | | √ | 30.7 | **11.0** | 30.1 | 41.1 | 7.5 | 40.2 |
| 5 | √ | √ | √ | | **32.9** | 11.5 | **32.1** | **45.3** | 7.2 | **44.3** |

in scenarios involving longer instructions and complex trajectories, demonstrating our ability to handle challenging long-term navigation tasks.

## 4.3 Ablation Studies

In this section, we conduct detailed ablation experiments to evaluate the effectiveness of each component proposed in our Loc4Plan, including the block-aware spatial locating (BAL) module and spatial-aware action planning (SAP) module.

**Effectiveness of the BAL and SAP module**. We first investigate the effectiveness of our overall design in Table 2. Row #1 presents the performance derived from our baseline approach. In this configuration, the agent's state representation relies exclusively on factors such as junction type, current heading angle, and visual cues. Instruction representation is attained through the average pooling of token embeddings. The results presented in Table 2 indicate that both the BAL and SAP modules contribute significantly to performance improvement. Specifically, the results delineated in rows #1 and #2 exhibit a promising enhancement in performance with the integration of the BAL module compared to the baseline. Furthermore, the inclusion of the SAP module, as evidenced in rows #1 and #3, yields notable improvements, enhancing the TC metric by 6.3% and 2.5% on map2seq and Touchdown, respectively.

**Investigation of the BAL module**. The BAL module is proposed to establish spatial positioning, which indicates its relative position within the observation field at the block level. Here, we investigate the components within the BAL module in Table 3. In addition to incorporating the current turning angle $g_t^c$, we innovatively introduced the long-term turning angle $g_t^l$ (Eq. (1)) for spatial perception. By separately comparing #1, #2 with the full model (#5), we can find that adding both the current and long-term turning angles results in improvements, validating their effectiveness. By comparing the results without supervision of the block process score (Eq. (5)) (#3) and with supervision (#5), we find that the inclusion of this supervision leads to a significant improvement of 4.3% and 2.6% in the TC metric on the map2seq and Touchdown datasets, respectively.

**Table 4: Ablation study of the perception length of long-term turning angle on test set in seen scenario.. The entries marked in gray indicate the default setting.**

| ID | Value of K | Touchdown | | | map2seq | | |
|----|------------|-----------|---|---|---------|---|---|
| | | TC↑ | SPD↓ | SED↑ | TC↑ | SPD↓ | SED↑ |
| 1 | 1 | 30.0 | 11.7 | 29.3 | 41.8 | 7.0 | 41.1 |
| 2 | 2 | 29.5 | 11.8 | 28.9 | 42.4 | **7.1** | 41.6 |
| 3 | 3 | **32.9** | 11.5 | **32.1** | **45.3** | 7.2 | **44.3** |
| 4 | 4 | 30.8 | 11.3 | 30.2 | 40.9 | 7.1 | 40.1 |
| 5 | 5 | 31.2 | **10.8** | 30.7 | 42.0 | 7.2 | 41.1 |

**Table 5: Ablation study of the spatial information usage in spatial-aware action planning module on test seen scenario.**

| ID | Model | Touchdown | | | map2seq | | |
|----|-------|-----------|---|---|---------|---|---|
| | | TC↑ | SPD↓ | SED↑ | TC↑ | SPD↓ | SED↑ |
| 1 | W/o spatial info | 32.4 | **11.4** | 31.7 | 42.3 | 7.3 | 41.4 |
| 2 | Full model | **32.9** | 11.5 | **32.1** | **45.3** | **7.2** | **44.3** |

**Table 6: Ablation studies of the submodules of the hierarchical semantic association on test set in seen scenario.**

| ID | Token | Sentence | $L_{HSA}$ | Touchdown | | | map2seq | | |
|----|-------|----------|-----------|-----------|---|---|---------|---|---|
| | | | | TC↑ | SPD↓ | SED↑ | TC↑ | SPD↓ | SED↑ |
| 1 | √ | | | 30.4 | 11.6 | 29.6 | 42.1 | 7.6 | 41.3 |
| 2 | | √ | √ | 28.7 | 12.2 | 27.5 | 41.4 | 7.5 | 40.6 |
| 3 | √ | √ | | 31.6 | **11.1** | 30.9 | 44.4 | 7.2 | 43.6 |
| 4 | √ | √ | √ | **32.9** | 11.5 | **32.1** | **45.3** | 7.2 | **44.3** |

Notably, in row #4, we replace the $L_{BAL}$ with the global-level locating loss $L_{GAL}$, which leverages global position signals to guide the spatial locating learning process. Our findings indicate the superiority of the block-level locating, given the agent's visual observation perception constrained within a localized region. To further investigate the effectiveness of perception length of long-term turning angle for the BAL module, we conducted experiments by varying the value of $K$ (Eq. (2)) in our model. The corresponding results are presented in Table4. Specifically, $K = 0$ implies that the model only utilizes the turning angle of the current node, while increasing values of $K$ expand the model's perception range to encompass a longer history of turning angles. The result shows that the agent achieves optimal performance when $K = 3$. This phenomenon indicates that a proper perception length of long-term turning angle is mandatory to achieve better performance.

**Investigation of the SAP module**. To investigate the impact of spatial information obtained from the BAL module on the action planning stage, we first show the results without using spatial information in the SAP module in table 5. Specifically, "W/o spatial info" means using the state representation $o_t$ to replace the spatial-aware feature $o_t^p$. Clearly, with the absence of spatial locating information, the agent's performance decreases on both datasets.

Moreover, to investigate the impact of the components involved in hierarchical semantic association, we conducted a series of ablation experiments, with the results presented in Table 6. Specifically, the label "token" and "sentence" refer to the multi-head attention

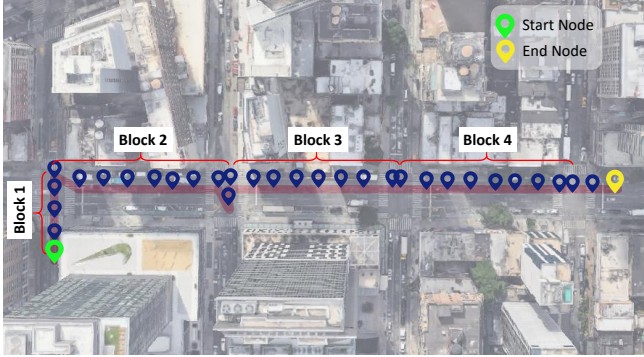

**Figure 4: Qualitative results of block process score (Eq. (5)) prediction in the BAL module. The green polyline represents the ground-truth block process across the entire trajectory, while the purple polyline depicts the corresponding predictions made by our method. The red brackets divided the navigation into multiple stages, with each stage encompassing nodes that belong to a single block.**

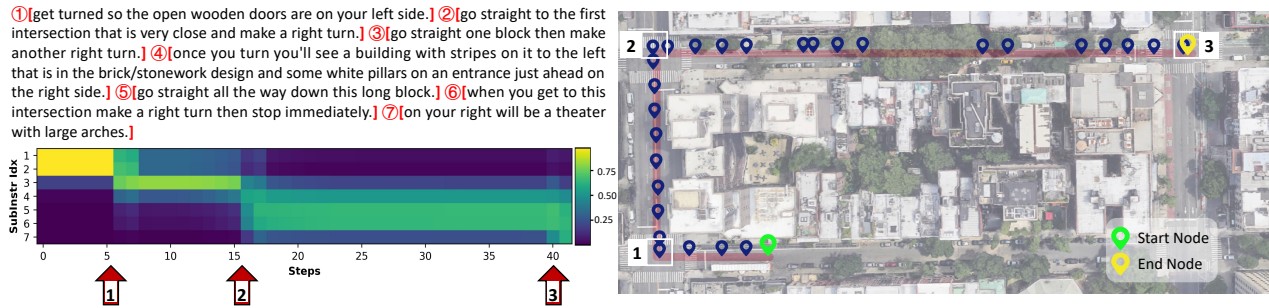

**Figure 5: Visualization of the sentence relevance scores (Eq. (8)) in HSA module. ①-⑦ indicate the range of each sentence in the instruction. The heatmap shows the degree of attention of our model to each sentence at each step. The red arrows pointed out three key navigation step, whose corresponding node positions in the scene graph are labeled with white box.**

on token level (Eq. (14)) and sentence level (Eq. (7)). "$L_{HSA}$" denotes the supervision on the sentence relevance score (Eq. (14)). The results reveal that using both sentence and token-level attention yields significantly better results than using either one alone, demonstrating their complementary roles in facilitating semantic association. Interestingly, the introduction of explicit supervision on sentence attention did not have as significant an impact as anticipated. Even without such supervision (#3), the model was able to achieve competitive navigation performance.

### 4.4 Visualization and Qualitative analysis

To intuitively show the effect of our method in spatial locating, a qualitative result is shown in Figure 4 that visualizes an example of the block progress score. It can be seen that our agent can track the progress of navigation across different blocks, which proves the ability in spatial localization. Moreover, Figure 5 presents a visualization of the agent's attention dynamics on the instruction sentences progress over multiple time steps. As shown in the heatmap of sentence relevance scores, our agent is able to focus on the relevant sentences within the instruction at different steps, especially exhibiting precise decision-making at several key steps.

## 5 CONCLUSION

In this work, drawing the inspiration from human navigation, we propose a "Locating before Planning (Loc4Plan)" framework. For the first time, this framework enables agents to locate their spatial position before planning a decision action based on corresponding guidance. The Loc4Plan framework comprises a block-aware spatial locating (BAL) module and a spatial-aware action planning (SAP) module. The BAL module is proposed to localize the spatial position on the block level, which enables the agent to be aware of its relative location position within the current block. The self-awareness of location ability developed in the BAL is beneficial for textual grounding, thereby facilitating further action planning. Therefore, we propose the SAP module for the planning process, which associates spatial-aware state representation with provided instructions in a hierarchical manner, promoting a comprehensive understanding of the provided instructions and make the action prediction. Benefiting from the ahead localization to the agent's position and comprehensive understanding of the provided instructions, our Loc4Plan achieves the new state-of-the-art for Touchdown and map2seq dataset on both seen and unseen scenarios.

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
