# OpenReview forum: "Loc4Plan: Locating Before Planning for Outdoor Vision and Language Navigation"
_acmmm.org/ACMMM/2024/Conference — MM2024 Oral_

### Official Review · Reviewer_r58c · 2024-05-23

**Rating:** 4
**Confidence:** 3

**Summary:**

They introduce a novel framework, Loc4Plan, designed to incorporate spatial perception for action planning in outdoor VLN tasks. The main idea behind Loc4Plan is to perform the spatial localization before planning a decision action based on corresponding guidance.

**Strengths:**

1. They introduced the agent location information into the action decision-making process.

2. The use of BAL and SAP modules significantly improves navigation performance.

**Limitations:**

1. They use LSTM and resnet as text and visual encoders respectively, which are too old. Why not use a transformer to extract features?

2. At line 414, they assume that $o_t^p$  accumulates the relevant information about spatial localization. I think this is unreasonable.

**Suitability:**

3

---

### Official Review · Reviewer_6TbU · 2024-05-23

**Rating:** 5
**Confidence:** 4

**Summary:**

This paper proposes a new framework named Loc4Plan consisting of a block-aware spatial locating (BAL) module and a spatial-aware action planning (SAP) module. The BAL module is used for localizing the spatial position on the block level and the SAP module can leverage this spatial information to make the action prediction.

**Strengths:**

- This design of the BAL module and the SAP module is creative.
- Numerous ablation studies conducted by the authors evaluate the effectiveness of various modules.
- This framework achieves the new State-Of-The-Art performance on two different datasets.

**Limitations:**

- The Vision-and-Language Navigation task is a sequence prediction task. However, the overview picture depicted in Figure 2 does not show this well. The illustration may lead to a misunderstanding that this task is a one-time prediction classification task.

**Suitability:**

3

---

### Official Review · Reviewer_dw5Y · 2024-05-25

**Rating:** 5
**Confidence:** 3

**Summary:**

The paper presents a method for outdoor vision-and-language navigation that is composed of two main modules block-aware spatial locating (BAL) and spatial-aware action planning (SAP). BAL identifies how far is the agent from the next intersection (where a turn can be made), while SAP grounds the instruction to the atomic actions exploiting the prediction returned by BAL.

**Strengths:**

The paper is engaging and well-written, presenting an interesting method for outdoor Vision-and-Language Navigation (VLN). The method is thoroughly described, ensuring the reproducibility of the code. The figures are well-integrated with the text, enhancing the comprehension of the method and providing valuable insights during the qualitative evaluation.

The method is thoughtfully designed, particularly with the introduction of two key modules for spatial localization and action planning. These modules significantly contribute to the overall experimental results.

The experimental evaluation is comprehensive, thoroughly examining all proposed modules. This extensive evaluation clearly demonstrates the improvement provided by each component of the architecture.

**Limitations:**

Regarding the limitations of the paper, my main concerns are related to doubts about the behavior of the model in some specific cases, in order to have an idea of the reasoning capabilities of the method.

How does it behave when the directions contained in the initial text instruction are not in order, e.g. "Go right at the second intersection after crossing the crossroad with the red building"?

I've seen that using the BAL module gives an improvement in the performance, but could the author explain better why predicting the distance from the next intersection is useful for action planning? The agent would need to reach the intersection anyway before making an eventual turn, and the information given by the observation captured at the intersection could be enough to make a decision. Why is using BAL better than forcing the agent to move directly at the next available intersection?

Could the authors elaborate on how the BAL output is beneficial for textual grounding?

What is the computational complexity of the model compared to the competitors?

The related work section on VLN could be expanded, for example, some recent work is missing:
[1]: Chen, S., Guhur, P. L., Tapaswi, M., Schmid, C., & Laptev, I., Learning from unlabeled 3d environments for vision-and-language navigation. ECCV 2022
[2]: Wang, X., Wang, W., Shao, J., & Yang, Y., Lana: A language-capable navigator for instruction following and generation. CVPR 2023
[3]: Rawal, N., Bigazzi, R., Baraldi, L., & Cucchiara, R., AIGeN: An Adversarial Approach for Instruction Generation in VLN. CVPRW 2024
[4]: Zhou, G., Hong, Y., & Wu, Q., Navgpt: Explicit reasoning in vision-and-language navigation with large language models. AAAI 2024

Some minor typos need to be fixed:
line 88 phrase-> phase
line 284 Priliminary -> Preliminary
line 449 plannin -> planning

**Suitability:**

3

---

### Official Review · Reviewer_F5dF · 2024-05-26

**Rating:** 5
**Confidence:** 3

**Summary:**

This work introduced a novel framework Loc4Plan for enhancing agent performance in outdoor vision and language navigation. The Loc4Plan framework contains block-aware spatial locating module and a spatial-aware action planning module for improving agent's grounding ability and action planning precision.

**Strengths:**

The paper is well-organized: The backgrond and motivation is clear. The function of each module is introduced in detail. The framework is clear to understand.

The experiment is convincing. The experiment is sufficient and analysized from quanititive and qualitative aspect.

**Limitations:**

Real-time performance of the system is not evaluated. Since this work focuses on outdoor vision and language navigation, computation efficiency and corner cases should be considered.

**Suitability:**

3

---

### Meta-Review · Area_Chair_kp7R · 2024-07-02

**Recommendation:** Accept (Oral)
**Confidence:** 5

**Metareview:**

There is consensus among reviewers that the paper is well-motivated, well-organized, and clearly written. The results presented in the paper are convincing and through ablations are presented as well. The author response addressed majority of the reviewers concern. I am recommending to accept the paper. However, the reviewers are advised to add the missing citations and discuss them in the paper.